# Comprehensive analysis of disulfidptosis-related genes in pulmonary hypertension through machine learning and immune infiltration: Spotlight on USP32 and ZNF655 as key regulators

Riken Chen[1☯], Dingyu Guo[2☯], Jiahua Pan[2☯], Lingpin Pang[2☯], Jie Sun[2], Qian Xian[2], Tao Huang[2], Junfen Cheng[1], Jihuang Huang[2], Xianbing Zeng[3], Guojun Yang[3], Shiyan Qi[3], Wenliang Chen[4]*, Xishi Sun[1,2]*

1 Respiratory and Critical Care Medicine, The Second Affiliated Hospital of Guangdong Medical University, Zhanjiang, Guangdong, China, 2 Emergency Medicine Center, Affiliated Hospital of Guangdong Medical University, Zhanjiang, Guangdong, China, 3 Lianjiang People's Hospital, Zhanjiang, Guangdong, China, 4 Scientific Research Center, The Second Affiliated Hospital of Guangdong Medical University, Zhanjiang, Guangdong, China

☯ These authors contributed equally to this work.
* febright@126.com (WC); 1097213689@qq.com (XS)

## Abstract

### Background

Disulfidptosis, a novel cellular death manner, has yet to be fully explored within the context of pulmonary arterial hypertension (PAH). This study aims to identify genes implicated in PAH that are involved in disulfidptosis.

### Method

Based on data from the GEO database, this study employed co-expression analysis, Weighted Gene Co-Expression Network Analysis (WGCNA), hub gene identification, and Gene Set Enrichment Analysis (GSEA) to uncover genes associated with PAH and disulfidptosis. Subsequent machine learning validation and functional GSEA further refined the identification of pivotal genes. The investigation extended to examining immune cell involvement via immune infiltration techniques and elucidates the hub genes' roles within ceRNA networks.

### Result

The integrative approach of co-expression analysis and WGCNA identified genes at the intersection of PAH and disulfidptosis. GSEA revealed their roles in essential biological processes and pathways, such as mRNA processing and cytoplasmic DNA sensing pathway. Prominently, USP32 and ZNF655 were identified as significant hub

**Data availability statement:** All relevant data are within the paper and its Supporting Information files.

**Funding:** This study was supported by the Health Development Promotion Project-Anesthesia and Critical Care Research Project (KM-20231120-01), Guangdong Medical Research Fund Project (A2024728, A2024723, B2025330, B2025378, B2025488), the Zhanjiang Science and Technology Research Project in 2022 (No: 2022A01197), the Science and Technology Development Special Fund Competitive Allocation Project of Zhanjiang City (No: 2021A05086), and the Guangdong Medical University Clinical and Basic Science Innovation Special Fund (No: GDMULCJC2024063, GDMULCJC2024064).

**Competing interests:** The authors have declared that no competing interests exist.

genes through machine learning analysis, demonstrating notable diagnostic potential across various datasets. Further, immune infiltration studies and ceRNA regulatory network construction revealed the intricate association between these genes and differential immune cell expression, alongside miRNA and lncRNA regulatory networks.

## Conclusions

This study elucidates the contributory role of USP32 and ZNF655 in the pathogenesis of PAH, making them as critical genes within the disulfidptosis pathway.

---

## 1. Introduction

Cardiovascular disease (CVD) stand as a leading cause of global mortality, prominently featuring idiopathic pulmonary arterial hypertension (PAH) among its most critical concerns. PAH is a rare multifactorial disease characterized by pulmonary vascular remodeling, progressive deterioration towards right heart failure, and an abnormal resistance to apoptosis [1]. Recent clinical delineations classify PAH within five subtypes of pulmonary hypertension (PH), identifying it as the most severe form, characterized by specific hemodynamic criteria including a mean pulmonary artery pressure (mPAP) exceeding 20 mmHg, a pulmonary artery wedge pressure (PAWP) of 15 mmHg or less, and a pulmonary vascular resistance (PVR) of 3 Wood units or more [2,3]. Patients with PAH typically undergo significant vascular remodeling, leading to vascular blockages [4]. Despite the identification of various biomarkers linked to vascular dysfunction, inflammation, myocardial stress, and tissue hypoxia, none have been specifically recognized as unique to PAH [1]. In genetics research, PH remains a challenging task despite its extensive study [5]. Current therapeutic strategies target three primary pathways: the nitric oxide (NO) pathway, the endothelin-1 (ET-1) pathway, and the prostacyclin (PGI) pathway, aiming to extend median survival time [6,7]. However, the overall patient survival rate after five years remains around only 57% to 59%, indicating the limited efficacy of existing treatments. Previous studies suggest immune disorders accounting for this gap in therapeutic outcomes, necessitating further investigations into PAH's molecular mechanisms and immune disorders to advance treatment approaches.

Disulfide, a compound stabilizing proteins during oxidative stress reactions, plays a pivotal role in preserving protein structures and ensuring their stability [8]. The discovery of 'disulfidptosis', a novel cell death pathway described by Liu and colleagues, marks a departure from traditional forms of programmed cell death (apoptosis, ferroptosis, and cuproptosis) [9]. Disulfidptosis primarily occurs under conditions of glucose starvation. Disulfidptosis arises when inhibited glucose metabolism leads to the upregulation of SLC7A11 facilitating increased cysteine uptake [10]. This, in turn, causes an excessive accumulation of disulfide in cells [11,12]. This accumulation precipitates a unique form of cell death, distinguished by its specific mechanism [9]. Given the potential link between genetic predispositions of vascular cells, inflammation, glycolysis metabolism alterations, and the pathogenesis of PAH [13], the

exploration of disulfidptosis within PAH presents a novel investigative avenue. However, to date, no studies have explicitly addressed the role of disulfidptosis-related genes in PAH.

MicroRNAs (miRNAs) represent a class of non-coding RNAs (ncRNAs), which are functional RNA molecules that are not translated into proteins [14]. Long non-coding RNAs (lncRNAs) are a class of transcripts exceeding 200 nucleotides in length that lack protein-coding capacity but play pivotal roles in gene regulation, biological processes, and diverse diseases [15]. Current research has established that both miRNAs and lncRNAs serve as critical regulators in PAH, modulating processes such as vascular remodeling, inflammation, and cellular proliferation through post-transcriptional gene silencing or competitive endogenous RNA (ceRNA) mechanisms [16,17]. Recent studies have revealed their regulatory roles in core pathogenic mechanisms of PAH, including Wnt signaling pathway activation and immune dysregulation [18,19]. For instance, lncRNAs can function as ceRNA that sequester miRNAs, thereby relieving their inhibitory effects on PAH-associated mRNAs [17].

This study aimed to illustrate the role of disulfidptosis-related genes in PAH by analyzing gene expression profiles from GSE15197 and GSE113439 databases. Through WGCNA, and subsequent Pearson correlation analysis, we identified crucial gene modules and hub genes using LASSO, random forest, and SVM-RFE algorithms. The association between these genes and immune cell activity was further examined through GSEA and single-sample GSEA (ssGSEA), culminating in the construction of a ceRNA network. This multifaceted approach not only aims to deepen our understanding of PAH at the molecular level but also to highlight potential therapeutic targets within its complex pathophysiological landscape.

## 2. Materials and methods

### 2.1. GEO dataset collection and data preprocessing

This study conducted data mining of PH and disulfidptosis-related genes using the GEO database (http://www.ncbi.nlm.nih.gov/geo/). We retrieved two raw high-throughput functional genomics datasets (GSE15197 [20] and GSE113439 [21]) related to PH. The GSE15197 dataset (GPL6480 platform) served as the experimental cohort, containing 26 PH cases (18 patients with PAH and 8 with secondary PH along with 13 normal controls obtained from fresh-frozen lung tissue specimens. The validation cohort comprised the GSE113439 dataset (GPL6244 platform), which included 15 PH cases (6 idiopathic PAH, 4 PAH secondary to connective tissue disease [CTD], 4 PAH secondary to congenital heart disease [CHD], and 1 chronic thromboembolic pulmonary hypertension [CTEPH]) and 11 normal controls acquired from tumor-adjacent lung tissues during surgical resection. We identified 10 genes associated with disulfidptosis for analysis within the GSE15197 dataset, including GYS1, NDUFS1, OXSM, LRPPRC, NDUFA11, NUBPL, NCKAP1, RPN1, SLC3A2, SLC7A11 [9], for analysis within the GSE15197 dataset. Selection criteria included human samples, inclusion of both pulmonary hypertension and control lung tissues, a sample size exceeding 15, and publicly available data.

### 2.2. Co -expression analysis

Utilizing the "Limma" package within R software, we performed correlation tests between disulfidptosis genes and the GSE15197 dataset (COR > 0.4, P-VALUE < 0.001). This approach facilitated the identification of the mRNA gene expression spectrum relevant to both pulmonary hypertension and disulfidptosis, visually represented through Sankey diagrams.

### 2.3. WGCNA and hub genes identification

We employed "WGCNA," an R software package, to construct a co-expression network for mRNA genes associated with pulmonary hypertension and disulfidptosis. Following the establishment of a scale-free network, optimal soft-thresholding, adjacency, and topological overlapping matrices (TOM) were determined. Dynamic module identification and Pearson correlation analyses were applied to discern module-gene relationships, with genes exhibiting a gene significance (GS) > 0.5 and module membership (MM) > 0.8 identified as core module genes.

## 2.4. Enrichment analysis

The "ClusterProfiler" package in R facilitated GO and KEGG pathway enrichment analyses for genes within significant modules, emphasizing biological processes (BP), molecular functions (MF), and cellular components (CC). Top results, based on the smallest P-values, were visually represented for both GO (top 10 results) and KEGG pathways (top 20 results with $P < 0.05$).

## 2.5. Machine learning (ML)-based hub gene screening and verification

The core genes were subjected to Lasso logistic regression, Random Forest (RF), and Support Vector Machine Recursive Feature Elimination (SVM-RFE) analyses to identify hub genes. The effectiveness of these hub genes was quantitatively assessed by calculating the area under the receiver operating characteristic curve (ROC-AUC) using the PROC package in R. A significance threshold of $P < 0.05$ was employed.

## 2.6. Single-based collection of rich set analysis (GSEA)

GSEA was conducted on individual hub genes to explore potential pathways and mechanisms implicated in pulmonary hypertension. Using the "ClusterProfiler" package in R, samples were divided into high- and low-expression groups based on median gene expression for comprehensive enrichment analysis.

## 2.7. Construction of the ceRNA regulatory network

To construct the ceRNA network for USP32 and ZNF655, we first predicted miRNA-target interactions using three bioinformatic algorithms: miRanda (http://www.microrna.org/), miRDB (http://mirdb.org/), and TargetScan (http://www.targetscan.org/). Stringent thresholds were applied (context score $< -0.2$ and conservation score $> 0.8$) to ensure prediction reliability. Subsequently, we identified lncRNAs capable of binding to these miRNAs using the spongeScan database (http://spongescan.rc.ufl.edu/), retaining only interactions with ≥2 predicted binding sites to ensure biological relevance. Finally, the ceRNA network was visualized using Cytoscape (v3.9.1), with nodes representing mRNAs, miRNAs, and lncRNAs, while edges denoted predicted interaction relationships.

## 3. Results

### 3.1. WGCNA analysis

Co-expression analysis identified 2,531 genes associated with pulmonary hypertension and disulfidptosis. Visualization using the 'ggalluvial' package in R software facilitated the creation of a Sankey diagram (Fig 1). These genes underwent WGCNA, setting a soft threshold of 19 to construct a weighted co-expression network, achieving an average connectivity near zero and an R2 value of 0.89 (Fig 2). Dynamic module identification isolated three significant modules—green, yellow, and gray. The green module comprises 1,271 genes, while the yellow module contains 950 genes. The gray module consists of 310 genes and is considered non-functional (The gray module typically includes genes that were not assigned to any other modules during WGCNA analysis. These genes may lack significant co-expression patterns or their expression profiles may differ from those in other modules). Hierarchical clustering further identified the green module as particularly relevant to disease pathology (Correlation = 0.6, p = 6e-05). Subsequently, using threshold criteria of Gene Significance (GS) > 0.5 and Module Membership (MM) > 0.8, we identified 246 highly connected hub genes exhibiting an inverse correlation with PAH within the green module. As shown in Fig 2E, there is a significant correlation(r = 0.31, p = 1e-29) between MM and GS, indicating a strong association between genes in the green module and gene significance.

### 3.2. Enrichment analysis

GO and KEGG analyses of the green module's core genes yielded significant insights (Fig 3). GO analysis (Fig 3A) highlighted enrichment in biological processes like mRNA processing and cell catabolism, and cellular components such

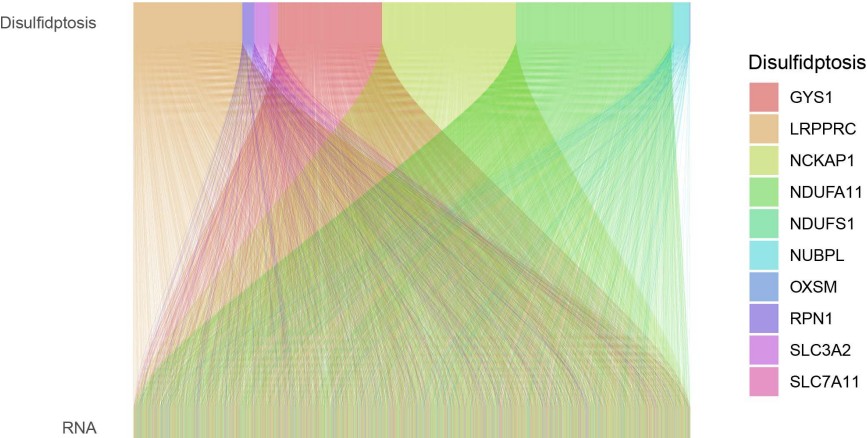

**Fig 1. Sankey Diagram in Co-expression Analysis.** Through co-expression analysis of disulfidptosis-related genes and pulmonary arterial hyperten-sion (PAH) gene expression profiles, we identified genes associated with disulfidptosis. A total of 5,656 genes were screened, including:1,055 genes co-expressed with the disulfidptosis gene GYS1;1,105 genes co-expressed with LRPPRC;1,360 genes co-expressed with NCKAP1;1,577 genes co-expressed with NDUFA11;29 genes co-expressed with NDUFS1;158 genes co-expressed with NUBPL;8 genes co-expressed with OXSM;122 genes co-expressed with RPN1;155 genes co-expressed with SLC3A2;87 genes co-expressed with SLC7A11.

as nuclear spots and protein complexes. Molecular functions showed enrichment in RNA binding and catalytic activities. KEGG pathway analysis (Fig 3B) revealed involvement in the cytoplasmic DNA sensing pathway, RNA degradation, and EB virus infection, glycosylphosphatidylinositol (GPI) anchored biosynthesis, and splicing.

### 3.3. Hub genes identification and validation

Applying LASSO, Random Forest, and SVM-RFE algorithms to 246 potential genes from the green module identified significant hub genes (Fig 4). LASSO regression isolated eight genes, including AKR7A2P1, AKR7A3, ATG3, RANBP6, TRAPPC9, TTLL12, USP32 and ZNF655, after 10-fold cross-validation (Fig 4A and 4B). SVM-RFE selected 11 hub genes (USP32, ZNF655, AHR, PNRC2, ERBB2IP, ZNF687, ZC3H15, PSMD12, YTHDF3, CPNE8, and ALKBH4) for their minimal error rates and optimal accuracy (Fig 4C and 4D), while Random Forest highlighted five disease-related genes, including ZNF655, AKR7A2P1, USP32, YTHDF3, and PNRC2 (Fig 4E and 4F). The intersection of these analy-ses confirmed USP32 and ZNF655 as critical hub genes (Fig 4G). ROC analysis validated their diagnostic effectiveness, with USP32 and ZNF655 exhibiting ROC values above 0.75 in both the experimental and validation datasets, underscor-ing their significance in pulmonary hypertension diagnosis (Fig 5). Therefore, we selected these two genes for further analysis.

### 3.4. Prediction of the potential pathway and mechanism function of hub genes

Finally, we used GSEA to predict the potential pathways and mechanism functions of the hub genes in the GSE15197 data-set. Fig 6 displays the top five most enriched GO functions and KEGG pathways in both the high-expression group (where hub genes are highly expressed in pulmonary hypertension patients from the GSE15197 dataset) and the low-expression group (where hub genes show low expression in pulmonary hypertension patients from the GSE15197 dataset).

**3.4.1. Top five most significantly enriched GO functions and KEGG pathways in the high-expression group.** In the high-expression group, USP32 was enriched in gene functions linked to cell response to biological stimulation, lymphocyte-mediated immunity, response to bacterial-derived molecules, chromosome region, and spindle (Fig 6A). USP32 was enriched in KEGG pathways including asthma, oocyte meiosis, p53 signaling pathway, terpenoid main chain

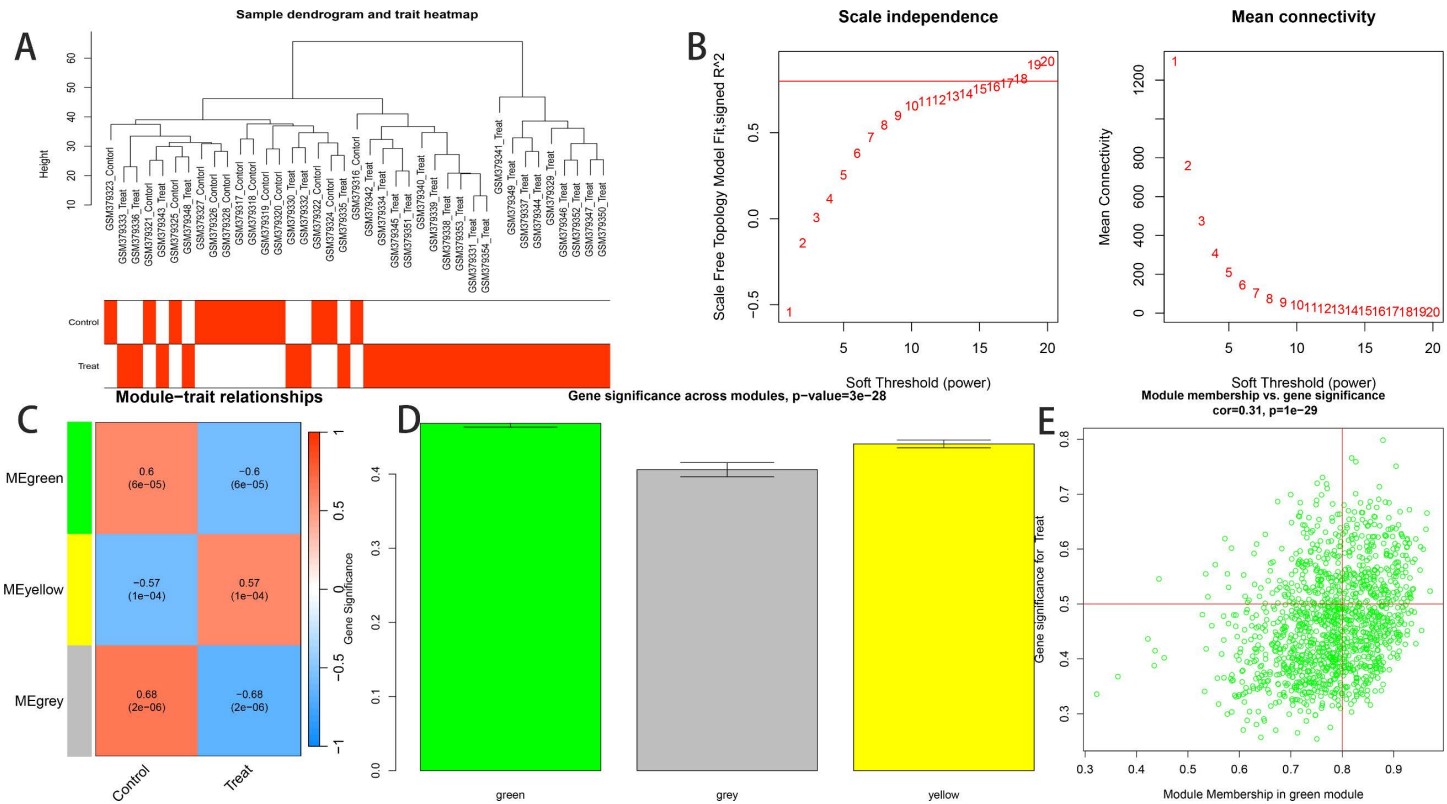

**Fig 2. Weighted Gene Co-expression Network Analysis (WGCNA) and Hub Genes Identification.** (A)shows "Sample Dendrogram and Trait Heatmap". This figure illustrates sample similarity and phenotypic classification. The dendrogram (generated via hierarchical clustering of gene expression data) reveals relationships among samples, with shorter branches indicating higher expression similarity. The trait heatmap uses color-coding: blue for the Control group and red for the Pulmonary Arterial Hypertension (PAH/Treat) group. (B)shows "Scale Independence Plot and Mean Connectivity Plot". The left panel displays the scale-free topology model fit (R²) across different soft-thresholding powers, while the right panel shows mean connectivity. The red line (R²=0.89) indicates that a soft-thresholding power of 18 was selected as optimal for constructing a biologically relevant co-expression network. (C)shows "Heat Map of Module-Trait Relationships". This heatmap depicts correlations between module eigengenes (y-axis: MEgreen, MEyellow, MEgrey) and phenotypic traits (x-axis: Control vs. PAH). Color intensity reflects correlation strength—dark yellow for strong positive, dark blue for strong negative—with p-values indicating significance. (D)shows "Gene Significance". The histogram presents the distribution of gene significance values (x-axis) across genes (y-axis). The green module shows the highest significance, suggesting its strongest association with PAH. (G)shows "Gene Significance Scatter Plot of the Green Module". This plot examines the relationship between module membership (x-axis) and gene significance (y-axis) in the green module, highlighting genes most strongly linked to PAH while maintaining high intramodular connectivity.

biosynthesis, and tryptophan metabolism (Fig 6B). ZNF655 was enriched in axoneme assembly, cilium movement, cilium or flagellum-dependent cell motility, microtubule bundle formation, axoneme, and other gene functions (Fig 6C). ZNF655 was enriched in KEGG pathways including drug Metabolism of cytochrome p450, drug metabolism of other enzymes, oocyte Meiosis, p53 Signaling pathway, and retinol metabolism (Fig 6D).

### 3.4.2. Top five most significantly enriched GO functions and KEGG pathways in the low-expression group.

In the low-expression group, USP32 was enriched in gene functions such as anterior-posterior pattern specification, canonical Wnt signaling pathway, embryonic organ development, gland development, and pattern specification process (Fig 6E). USP32 was enriched in KEGG pathways including basal cell carcinoma, cardiac muscle contraction, focal adhesion, maturation-onset diabetes of the young, and Wnt signaling pathway (Fig 6F). ZNF655 was involved in anterior-posterior pattern specification, regulation of trans-synaptic signaling, G protein-coupled receptor activity, and peptide

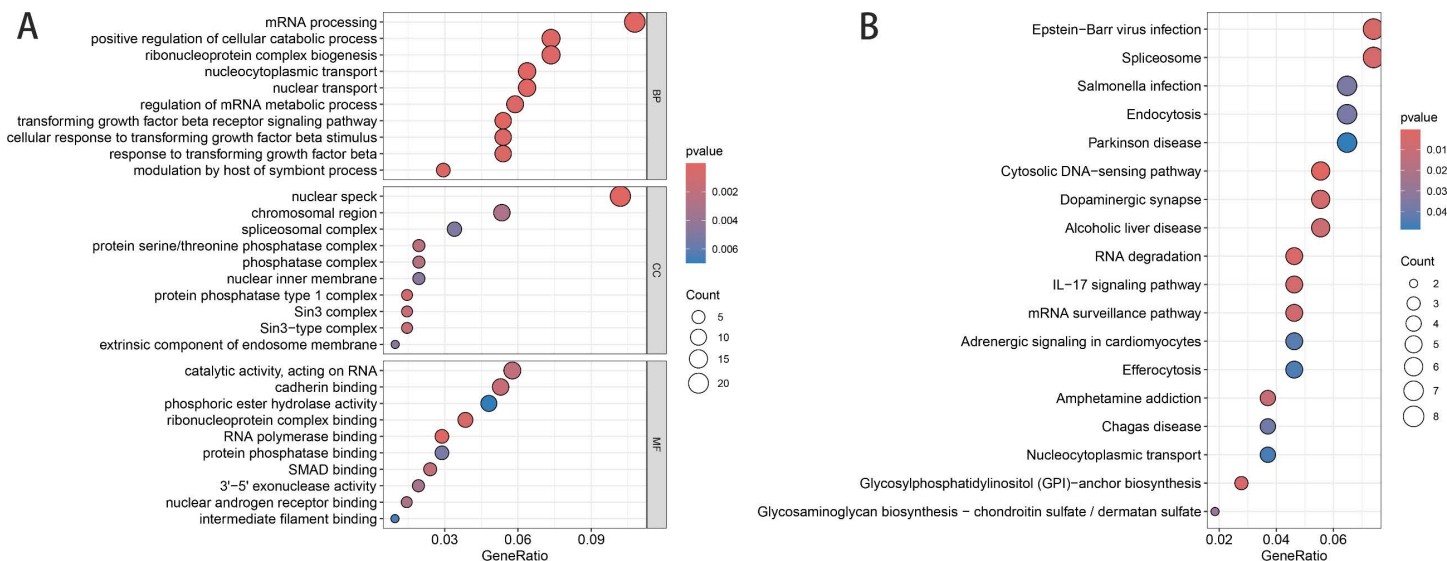

**Fig 3. Enrichment Analysis Results.** Fig 3 presents the Gene Ontology (GO) and Kyoto Encyclopedia of Genes and Genomes (KEGG) enrichment analyses performed on the core genes identified in the green module. (A) shows the GO enrichment analysis, covering BP, CC, and MF. The bubble plot displays the top five significantly enriched terms, where the bubble size corresponds to the number of genes involved (larger bubbles represent higher gene counts) and the color gradient reflects the adjusted p-value significance (more intense red indicates greater statistical significance). (B) shows the KEGG pathway enrichment analysis, featuring the top 20 enriched pathways in a similar bubble plot format. The visualization highlights the most relevant metabolic and signaling pathways, with bubble size and color coding maintaining the same representation as in panel A for consistent interpretation.

receptor activity (Fig 6G). ZNF655 was enriched in KEGG pathways including basal cell carcinoma, calcium signaling pathway, cardiac muscle contraction, juvenile diabetes mellitus, and neuroactive ligand receptor interaction (Fig 6H).

### 3.5. Immune infiltration analysis

For further study, ssGSEA was used to compare immune cell distributions between control and pulmonary hypertension samples within the GSE15197 dataset. In the heat map, the distribution of 28 immune cells in each sample of GSE15197 dataset was shown. The Fiddle diagram (Fig 7) revealed significant differences in several immune cell types, including activated CD4 T cells, CD56 bright natural killer cells, eosinophils, gamma delta T cells, immature dendritic cells, mast cells, natural killer T cells, natural killer cells, neutrophils, regulatory T cells, T follicular helper cells, and effector memory CD8 T cells, suggesting their involvement in PAH progression. Immunocorrelation analysis indicated positive correlations of USP32 with dendritic cells and T cells (p<0.001), memory B cell (p<0.05), activated CD4 T cell (p<0.05), and negative correlations with specific T helper cells (p<0.05), central memory CD8 T cell (p<0.05), and CD56dim natural killer cell (p<0.05). ZNF655 showed similar immune correlations, emphasizing the genes' roles in immune modulation in PAH.

### 3.6. Construction of a ceRNA regulatory network

We constructed ceRNA networks for USP32 (Fig 8) and ZNF655 (Fig 9), predicting related miRNAs and lncRNAs using databases like miRanda, miRDB, and TargetScan, obtaining 73 miRNAs. This approach yielded complex networks, illustrating the intricate regulatory relationships involving USP32 and ZNF655 within pulmonary hypertension pathology. Then, the above miRNA-related lncRNAs were predicted by spongScan database, and finally only 25 miRNAs predicted 70 related lncRNAs. Similarly, 59 miRNAs and 82 lncRNAs were predicted related to ZNF655. The lncRNA-miRNA-mRNA ceRNA network was then constructed using Cytoscape to obtain a visual map.

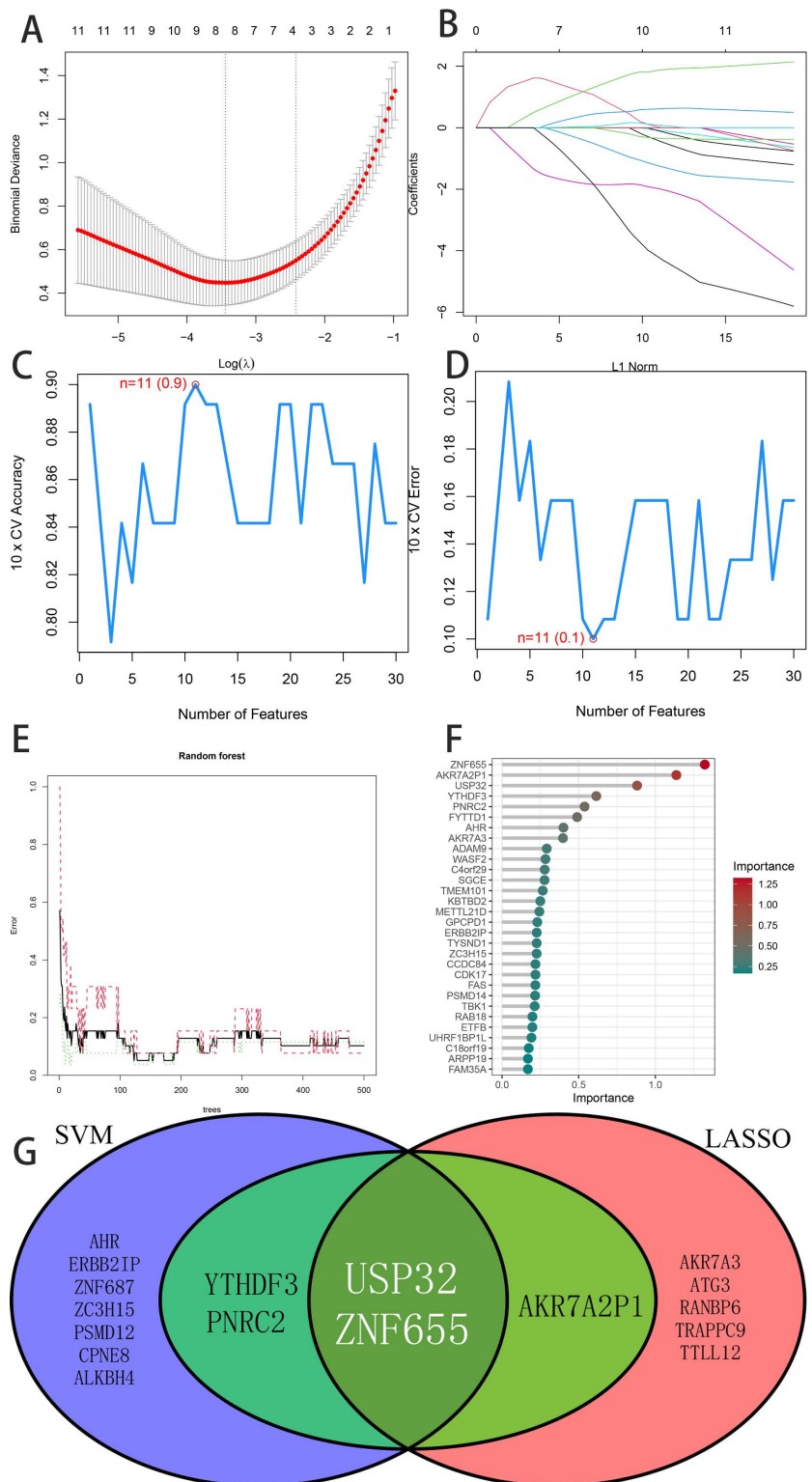

**Fig 4. Hub Genes selection Results.** (A) and (B) show that the Least Absolute Shrinkage and Selection Operator(LASSO) regression algorithm was applied for feature gene selection, with the regularization parameter λ used for covariate selection and dimensionality reduction. (C) and (D) show that feature genes were screened using the Support Vector Machine-Recursive Feature Elimination(SVM-RFE) algorithm, an iterative approach that ranks

and eliminates the least important features based on classifier performance. (E) and (F) show that the Random Forest(RF) algorithm was employed for feature gene selection, leveraging its built-in feature importance scoring to identify the most relevant genes. (G) shows a Venn diagram illustrating the overlapping key feature genes identified by LASSO, SVM-RFE, and RF, highlighting consensus biomarkers across different selection methods. The genes selected by Lasso include AKR7A2P1, AKR7A3, ATG3, RANBP6, TRAPPC9, TTLL1, USP32, and ZNF655. The genes selected by the SVM-RFE algorithm include USP32, ZNF655, AHR, PNRC2, ERBB2IP, ZNF687, ZC3H15, PSMD12, YTHDF3, CPNE8, and ALKBH4. The genes selected by the RF algorithm include ZNF655, AKR7A2P1, USP32, YTHDF3, and PNRC2.

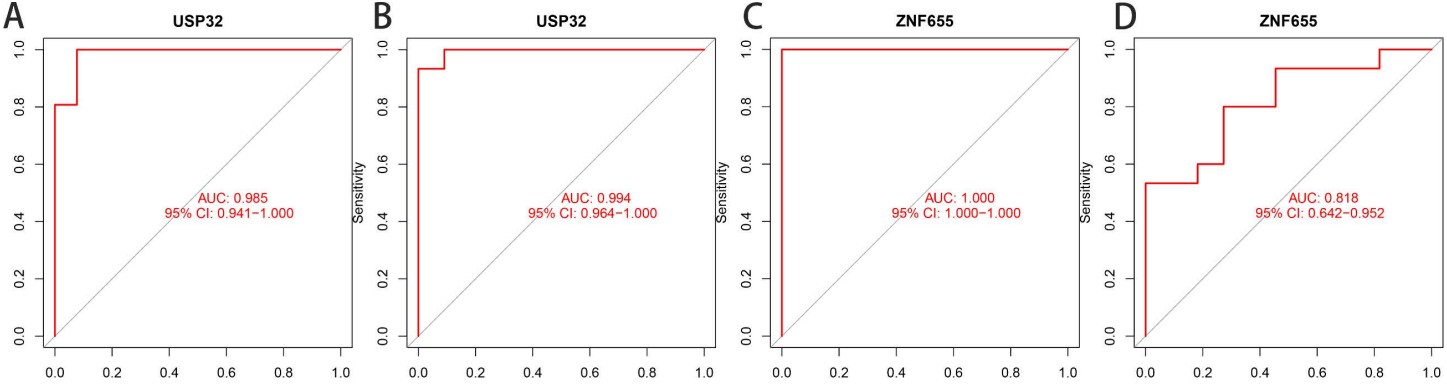

**Fig 5. Area Under the Receiver Operating Characteristic Curve(AUC-ROC) Results.** (A) shows the ROC curve of gene USP32 in the training set GSE15197 (GPL6480), demonstrating its diagnostic performance. (B) shows the ROC curve of gene USP32 in the independent validation set GSE113439 (GPL6244), confirming its robustness. (C) shows the ROC curve of gene ZNF655 in the training set GSE15197 (GPL6480), evaluating its classification efficacy. (D) shows the ROC curve of gene ZNF655 in the validation set GSE113439 (GPL6244), further validating its predictive capability.

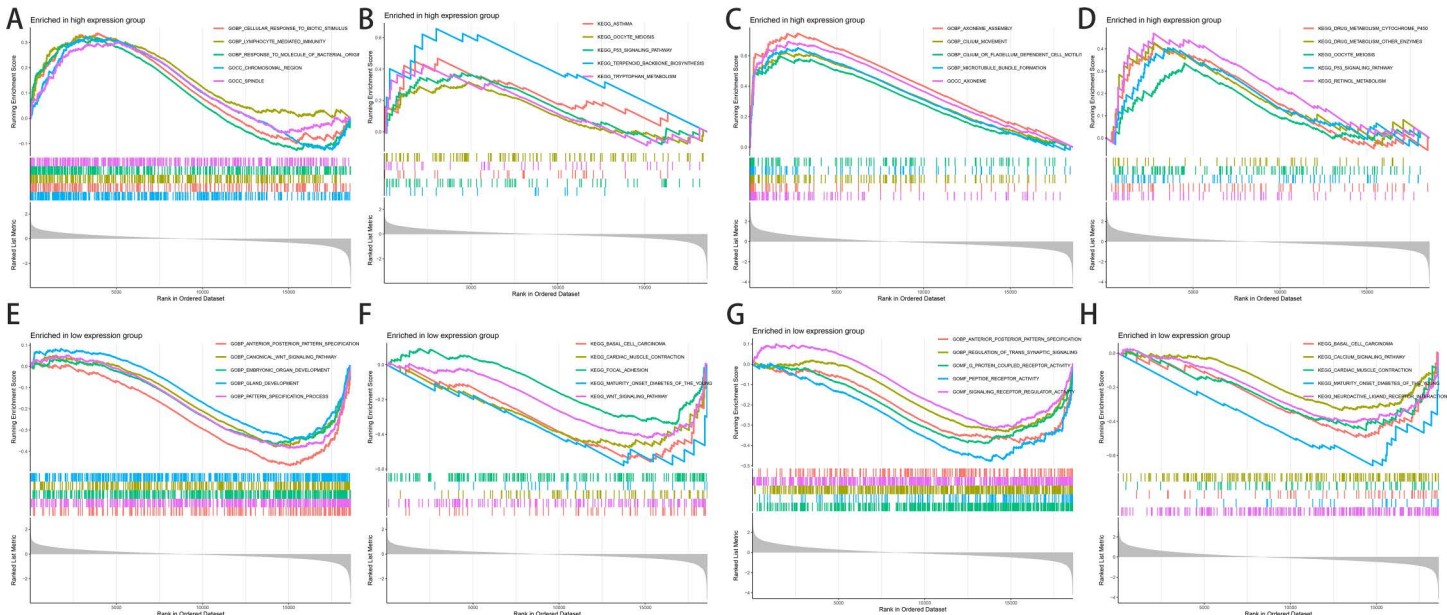

**Fig 6. GSEA Results.** Fig 6 shows the GSEA-predicted potential pathways and mechanistic functions of hub genes in the GSE15197 dataset. (A)and (B) show GO and KEGG enrichment analyses for the USP32 high-expression group, revealing its potential biological roles and associated pathways. (C) and (D) show GO and KEGG enrichment analyses for the USP32 low-expression group, highlighting distinct functional mechanisms compared to the high-expression group. (E) and (F) show GO and KEGG enrichment analyses for the ZNF655 high-expression group, identifying key biological processes and signaling pathways linked to its overexpression. (G) and (H) show GO and KEGG enrichment analyses for the ZNF655 low-expression group, uncovering differential pathway activities relative to high-expression conditions.

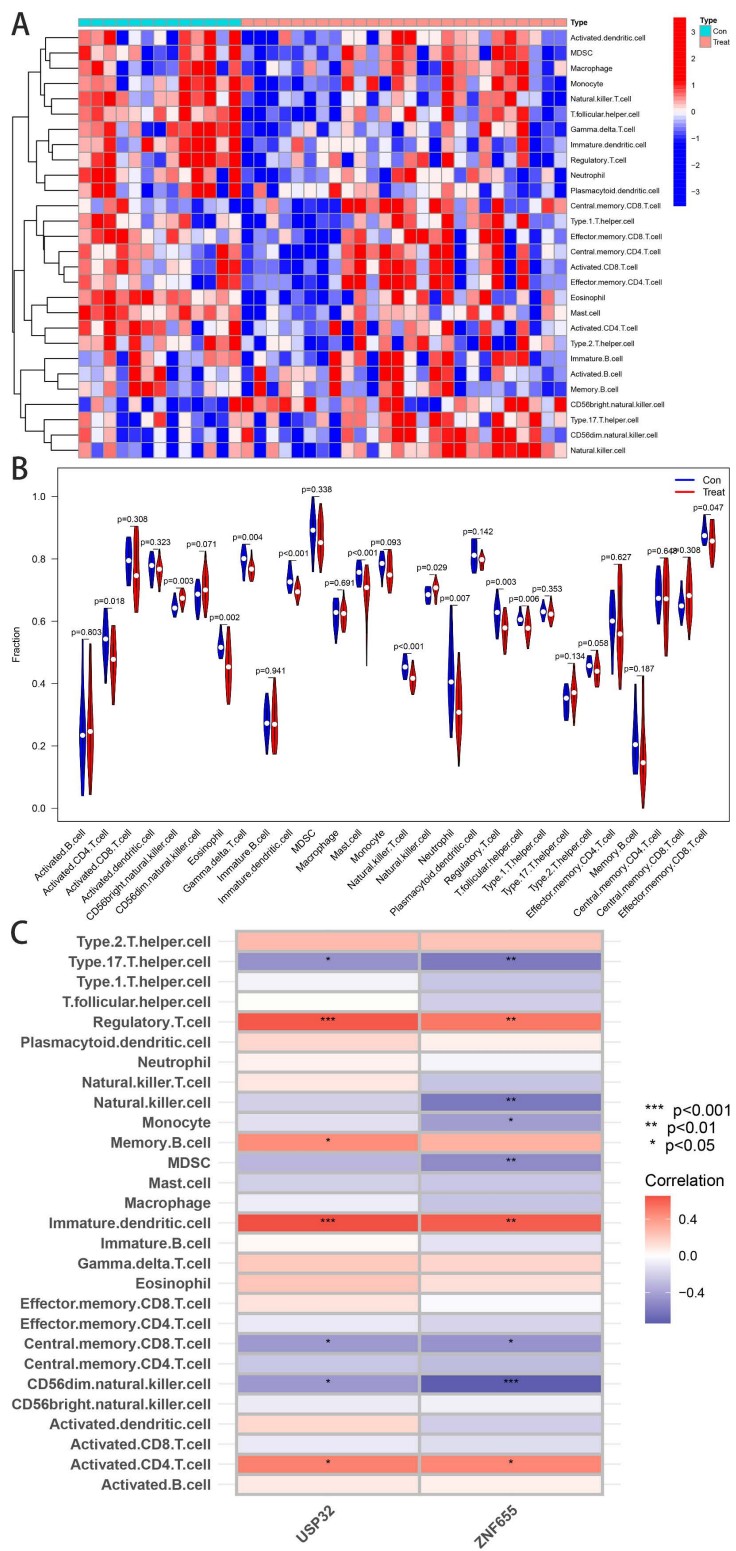

**Fig 7. The Violin Plot Obtained From Immune Infiltration Analysis.** (A) shows a heatmap depicting the differential distribution of 28 immune cell subtypes across study samples, highlighting distinct immune infiltration patterns between the PAH group and controls. (B) presents violin plots comparing the infiltration levels of 28 immune cell populations between control subjects and PAH patients, demonstrating significant differences in immune cell

abundance between groups. (C) displays box plots illustrating the expression differences of hub genes (USP32 and ZNF655) between control and PAH groups, revealing their potential roles in disease pathogenesis.

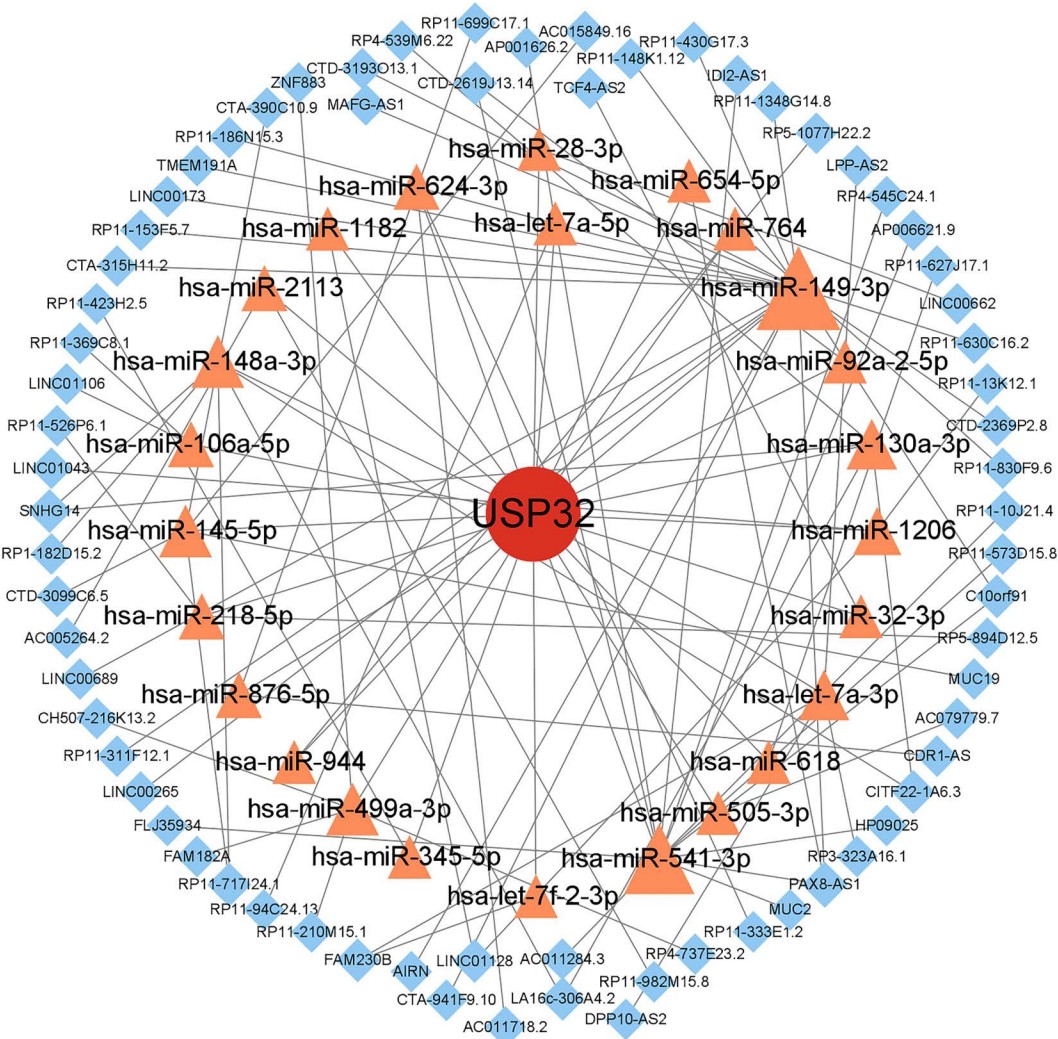

**Fig 8. ceRNA regulatory network diagram——USP32.** Fig 8 shows the ceRNA regulatory network diagram of miRNA and lncRNA related to USP32.

## 4. Discussion

PAH is a complex cardiovascular disorder characterized by progressive right heart failure and high mortality, with affected patients exhibiting significantly reduced life expectancy [22,23]. Despite the advent of numerous therapeutic strategies in recent years, contemporary treatments principally mitigate the consequences of disease rather than offering a cure [24,25]. Consequently, elucidating the molecular mechanisms underlying PAH progression, developing targeted therapies against novel pathways, and validating clinically actionable biomarkers represent urgent unmet needs to transform patient management. Recent studies have identified disulfide-dependent cell death, otherwise referred to as disulfide stress-induced cell death (DSCCD), as an additional factor contributing to pulmonary vascular remodeling in patients diagnosed

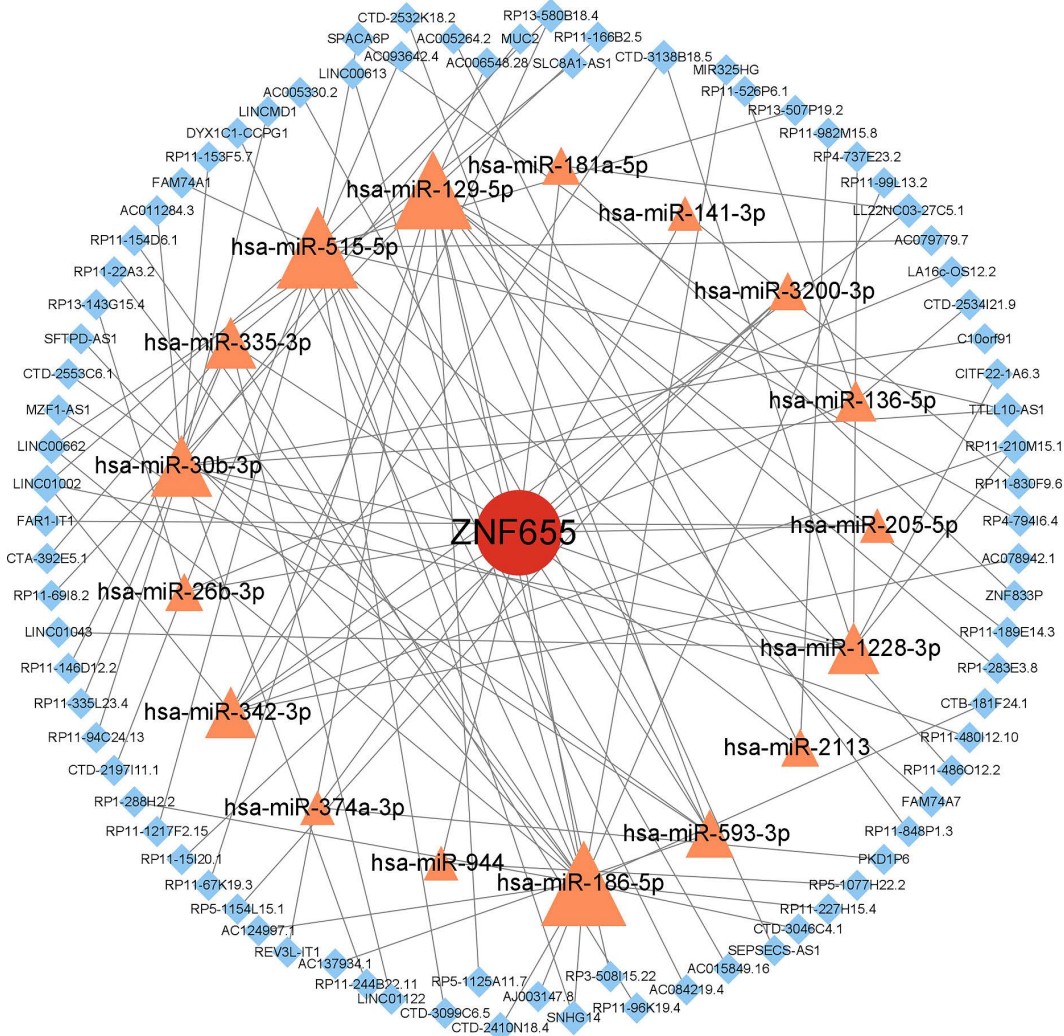

**Fig 9. ceRNA regulatory network diagram——ZNF655.** Fig 9 shows the ceRNA regulatory network diagram of miRNA and lncRNA related to ZNF655.

with PAH. This form of cell death is iron-dependent, and it has been shown to play a significant role in the development of this condition [26,27]. The objective of this study is to methodically examine the mechanistic associations between disulfide-mediated protein folding and PAH by employing comprehensive bioinformatics methodologies. This investigation seeks to elucidate the pathophysiological underpinnings that contribute to the progression of these diseases.

In the present study, we initially identified USP32 and ZNF655 as hub genes that were significantly upregulated in patients with PAH through a combination of WGCNA and machine learning approaches. WGCNA, a systems biology method that has gained significant popularity among researchers, offers an effective approach to comprehensively characterize the gene-gene association patterns present in microarray datasets. This method has been extensively employed in the realm of bioinformatics research, as evidenced by its numerous applications [28]. The present analysis indicates that the aforementioned amplification strategy is indeed effective in network-based discovery. This finding serves as a validation of the strategy's robustness and serves to substantiate the efficacy of the approach. ML is a pivotal subfield of artificial intelligence (AI) that has been widely utilized in various disciplines, with particularly prominent applications in

bioinformatics research [29]. In this study, we employed multiple machine learning approaches—including Lasso logistic regression, RF, and SVM-RFE—to analyze the green module identified through WGCNA, ultimately identifying hub genes with high network centrality. As demonstrated in previous studies, PAH is characterized by a multifactorial pathophysiology, which is typically accompanied by progressive right heart failure and abnormal pulmonary vascular remodeling [30]. Pulmonary vascular remodeling is a complex process influenced by multiple mechanisms, including excessive proliferation, apoptosis of vascular cells, and infiltration of inflammatory cells into the vasculature [31]. The broad category of cell death modalities comprises two primary forms: programmed cell death (PCD) and necrotic cell death, which includes the previously mentioned disulfide-dependent process. These mechanisms play pivotal roles in cellular homeostasis regulation [32,33]. USP32 encodes a member of the ubiquitin-specific protease (USP) family. Ubiquitination, a pivotal post-translational modification of intracellular proteins, plays a critical role in regulating their specific functions [34]. As indicated by the findings of preceding studies, USP32 functions as a pivotal regulator across a variety of pathways, thereby participating in diverse cellular biological processes. The present study demonstrates that the aforementioned pathway is functionally linked to a variety of biological processes, including cell cycle regulation, cell death, proliferation, invasion, migration, DNA replication, base excision repair, mismatch repair, and DNA damage response pathways. These processes are closely associated with the development of PAH [35,36]. ZNF655 is a member of the zinc-finger (ZNF) protein family. The ZNF protein family, the largest transcription factor family in the human genome, exhibits diverse molecular functions and participates in multiple cellular processes through distinct molecular mechanisms [37]. Distinct combinations of functional motifs within the ZNF protein regulate diverse biological processes, including development, differentiation, metabolism, and autophagy. These mechanisms correspond remarkably to the pathological vascular remodeling observed in PAH [38,39].

To further elucidate the gene expression data, we performed GSEA. This method entails the analysis of predefined gene sets to derive biologically relevant functions, a practice that has seen an uptick in recent studies [40]. In the USP32 high-expression group, the activation of lymphocyte-mediated immune responses has been demonstrated to play a pivotal role in the regulation of inflammation and the dysregulation of the cell cycle in patients with PAH. These mechanisms may be associated with the initiation and maintenance of immune-mediated inflammation during pulmonary vascular remodeling in patients with PAH [41–44]. A body of research has demonstrated that the systemic administration of human endothelial cell-cultured modified monocytes can prevent the development of PAH by stimulating innate immune lymphocytes. This finding suggests the potential role of these immune cells in PAH pathophysiology [45]. The observed congruence between these well-established findings and the subsequent results offers independent validation of the conclusions drawn. In the USP32 low-expression group, the reduced levels of USP32 may result in the suppression of the Wnt signaling pathway, thereby impairing vascular development and endothelial function. These mechanisms may be associated with vascular remodeling in patients with PAH. The veracity of our findings was corroborated by the study of Yuan K., et al. Their research demonstrated that a pivotal factor in the canonical Wnt signaling pathway modulates pulmonary endothelial cell-pericyte interactions, and the absence of this factor promotes the development of PAH by diminishing the viability of new blood vessels [46]. The findings of this investigation align with those of previous studies, which also provide substantiating evidence for our research on ZNF655 [47,48]. Of particular interest is the finding that the p53 signaling pathway was found to be significantly enriched in both USP32 and ZNF655 analyses. As demonstrated in previous experimental studies, p53 inactivation has been shown to trigger pulmonary hypertension and vascular remodeling. p53 dysregulation has been found to induce the upregulation of hypoxia-inducible factor-2α (HIF-2α), which in turn promotes endothelial-to-mesenchymal transition (EndMT). These processes have been identified as exacerbating factors in the pathogenesis of pulmonary hypertension. These findings establish p53 as playing a pivotal role in PAH development [49–51].

To further characterize the biological functions associated with this feature, we performed ssGSEA. A body of research has previously identified perivascular inflammation as a salient pathophysiological feature in PAH. A characteristic accumulation of diverse immune cells—including neutrophils, macrophages, dendritic cells, mast cells, T lymphocytes, and B lymphocytes—is

observed around pulmonary vessels in patients with PAH. This accumulation represents nearly the complete spectrum of inflammatory cell lineages [31,52,53]. In this study, we observed a positive correlation between USP32 expression and the degree of dendritic cell infiltration, as well as the extent of T-cell activation and memory B-cell abundance in the tissue samples. These findings suggest a potential role for USP32 in regulating adaptive immune responses during the progression of PAH. In contrast, the ZNF655 high-expression group exhibited significant associations with neutrophil and natural killer cell infiltration, implicating innate immune pathways in vascular inflammation and remodeling. These findings demonstrate that differential immune cell expression patterns and their specific genetic correlations may contribute to the development and progression of PAH, providing critical insights for further investigation into immune-mediated mechanisms in PAH pathogenesis.

Consequently, a ceRNA network was constructed. The USP32-centered ceRNA network comprised 25 microRNAs (including hsa-let-7a-5p and hsa-miR-145-5p) and 70 long non-coding RNAs (such as RP11-10J21.4 and MUC19), revealing extensive post-transcriptional regulatory potential in the pathogenesis of PAH. As indicated by prior research, members of the let-7 family, such as hsa-let-7a-5p, have been observed to exhibit significant upregulation in lung tissues of patients diagnosed with idiopathic pulmonary arterial hypertension (IPAH). In these tissues, these molecules have been shown to promote vascular remodeling through the targeted regulation of the Wnt/β-catenin pathway [54]. The ZNF655-associated ceRNA network comprised 17 microRNAs (including hsa-miR-186-5p and hsa-miR-26b-3p) and 82 long non-coding RNAs (such as RP11-830F9.6 and SNHG14), underscoring its extensive regulatory potential in pulmonary vascular homeostasis. It has been demonstrated that has-miR-181a-5p expression is reduced in PAH, and that this reduction contributes to the attenuation of inflammatory responses and vascular remodeling by targeting genes such as TNF-α. This, in turn, contributes to the pathogenesis of PAH [55]. From a mechanistic perspective, it is hypothesized that hsa-miR-205-5p functions as a suppressant of proliferation in pulmonary arterial smooth muscle cells (PASMCs) within the pulmonary arterial hypertension (PAH) context. This hypothesis posits that the suppression is achieved through the targeting of MICAL2, consequently impeding the downstream activation of ERK1/2 signaling [56]. These findings suggest that ZNF655 may regulate key PAH pathways through mechanisms involving microRNAs. The findings of the present study are consistent with those of previous research, which has demonstrated the critical role of non-coding RNA networks in pulmonary vascular remodeling.

## 5. Conclusion

In summary, our study demonstrates USP32 and ZNF655 as hub genes in the progression of PAH, involved in a variety of critical biological processes and signaling pathways, and closely related to the immune regulation in PAH. These findings not only deepen our understanding of the pathological mechanisms of PAH but may also shed a light for the development of new therapeutic targets and strategies. Future research should further investigate the specific mechanisms and roles of these hub genes in PAH pathogenesis, offering a promising direction for advancing our comprehension of PAH. While preliminary, the analysis presented here provides important clues for the ongoing study of PAH, potentially aiding in the development of more effective interventions for this challenging condition.

## Supporting information

**S1 Data. Code.**
(ZIP)

**S2 Data. Data.**
(ZIP)

## Author contributions

**Conceptualization:** Qian Xian, Riken Chen.

**Data curation:** Jiahua Pan, Shiyan Qi, Dingyu Guo.

Formal analysis: Jihuang Huang.

Funding acquisition: Riken Chen.

Investigation: Jihuang Huang, Xishi Sun, Junfen Cheng.

Methodology: Jiahua Pan, Tao Huang, Dingyu Guo, Xishi Sun.

Project administration: Lingpin Pang, Tao Huang, Jihuang Huang, Wenliang Chen.

Resources: Lingpin Pang, Jihuang Huang, Xishi Sun.

Software: Jie Sun, Guojun Yang.

Supervision: Jie Sun, Tao Huang, Xishi Sun, Junfen Cheng.

Validation: Qian Xian, Xianbing Zeng, Xishi Sun, Junfen Cheng.

Visualization: Qian Xian, Xishi Sun.

Writing – original draft: Dingyu Guo.

Writing – review & editing: Dingyu Guo, Xishi Sun, Riken Chen.

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
