## [Decision Letter · Decision Letter 0]

23 Mar 2025

Dear Dr. Guo,

Thank you for submitting your manuscript to PLOS ONE. After careful consideration, we feel that it has merit but does not fully meet PLOS ONE’s publication criteria as it currently stands. Therefore, we invite you to submit a revised version of the manuscript that addresses the points raised during the review process.

**ACADEMIC EDITOR:**

While the manuscript presents a valuable methodological contribution, both of the reviewers have suggested that a revision is needed to improve the visualization and enhance the practical impact of the work. Please revise the manuscript accordingly and incorporate the reviewers' suggestions.

We look forward to receiving your revised manuscript.

Kind regards,

Mahdi Roozbeh

Academic Editor

PLOS ONE

Journal Requirements:

This study was supported by the Health Development Promotion Project-Anesthesia and Critical Care Research Project (KM-20231120-01), Guangdong Medical Research Fund Project (A2024728、A2024723), the Zhanjiang Science and Technology Research Project in 2022 (No: 2022A01197), and the Science and Technology Development Special Fund Competitive Allocation Project of Zhanjiang City (No: 2021A05086).

4. We note that your Data Availability Statement is currently as follows: The datasets generated and/or analysed during the current study are available from the corresponding author on reasonable request.

Reviewers' comments:

Reviewer's Responses to Questions

**Comments to the Author**

1. Is the manuscript technically sound, and do the data support the conclusions?

Reviewer #1: Yes

Reviewer #2: Partly

2. Has the statistical analysis been performed appropriately and rigorously?

Reviewer #1: N/A

Reviewer #2: Yes

3. Have the authors made all data underlying the findings in their manuscript fully available?

Reviewer #1: Yes

Reviewer #2: No

4. Is the manuscript presented in an intelligible fashion and written in standard English?

Reviewer #1: No

Reviewer #2: Yes

Reviewer #1: The author reported an interesting study that utilized transcriptomic datasets to identify disulfidptosis-related genes involved in PAH. Below are my comments and suggestion for the manuscript:

1. For a better understanding of the study and datasets, I suggest providing a clear description and discussion of the nature of the datasets used.

2. The findings presented in Figure 1 require further elaboration.

3. Further elaboration is needed for each panel in Figure 2. What do the modules represent? Do they correspond to genes? How many genes are included in each panel? Why is the grey module considered insignificant?

4. Figure 3, Is there a correlation between the outcomes of GO enrichment analysis and KEGG enrichment analysis?

5. What criteria led to the selection of the green module over the yellow module for further analysis? Panel G, the list does not seem too extensive. Please provide the gene names instead of numbers.

6. Figure 5, please state the differences between (panel A & panel B) and (panel C & panel D).

7. The sentence on page 9/29, starting with “Figure 6 showed that the high...” until the end, is unclear. Please revise it for clarity.

8. Figure 7, the Fiddle diagram: What is the meaning of “Con” and “Treat”? What kind of differences exist between the cells? Are they related to cell number, gene expression, or other factors?

9. The discussion is lengthy and lacks a well-organized flow. I suggest focusing more on explaining the findings and linking them to the disease.

Reviewer #2: In this work, Chen and colleagues utilized GEO datasets to investigate the expression of disulfidptosis-related genes in arterial hypertension (PAH). After constructing gene modules with WGCNA, they performed GO and KEGG analyses of the module with increased significance to disease pathology. They applied LASSO, Random Forest, and SVM-RFE in parallel to narrow down the list of hub genes to a couple of overlapping genes (USP32 and ZNF655). They further used GSEA to determine potential pathways and mechanism functions of the hub genes, performed immune infiltration analysis to determine immune cell distributions between control and pulmonary hypertension groups, and built ceRNA regulatory network for USP32 and ZNF655 to predict related miRNA and lncRNA regulatory networks.

Major concerns:

The figures with higher resolution (text is not completely visible in figures 2, 6, 7, and 8, for example), and more detailed legends are necessary (including experiment descriptions, and how the graphs were generated). This is essential to fully visualize and understand the data, as well as follow the interpretation of the results.

The authors present interesting data produced through a well structured and logical analysis, yet limited by sample size and lack of experimental validation. USP32 and ZNF655 are involved in multiple cellular processes and allow for generating many hypotheses about their role in PAH, therefore, the relevance of their findings could be strengthen by focusing the discussion primarily on how the data connects with what is already well known about the cell and molecular biology of PAH (instead of tackling multiple connections not necessarily as robust).

Minor concerns:

In “Dynamic module identification isolated three significant modules—green, yellow, and gray—with the gray module deemed nonsignificant. Hierarchical clustering further identified the green module as particularly relevant to disease

pathology”, can you explain why gray module was deemed nonsignificant? And why is green module is relevant to disease pathology?

In Figure 2, groups are divided into ‘control’ and ‘treat’. Can you explain what “treat” means?

In Figure 4 legend, reference to C is missing.

In Figure 5, what is the difference between A and B, and C and D?

Duplicated sentences in discussion: “Furthermore, studies involving bovine subjects have revealed that the expression of olfactory receptors (ORs) is governed by MOR4, a member of the GPCR superfamily. MOR4 is primarily recognized as a binding site for the zinc finger (ZNF) transcription factor gene family, suggesting a link between GPCR activity and ZNF[41]. Furthermore, studies involving bovine subjects have revealed that the expression of olfactory receptors (ORs) is governed by MOR4, a member of the GPCR superfamily. MOR4 is primarily recognized as a binding site for the zinc finger (ZNF) transcription factor gene family, suggesting a link between GPCR activity and ZNF [42].”

**Do you want your identity to be public for this peer review?** For information about this choice, including consent withdrawal, please see our Privacy Policy

Reviewer #1: No

Reviewer #2: No

---

## [Author Response · Author response to Decision Letter 1]

6 May 2025

Hello, I'm glad you had a few times to review our work, and we've already made revisions to the manuscript and replied point by point. However, the raw data and code shares are not accessible to the website, so we've uploaded to "Other" and "Support Information" separately and asked you to let me know what to do next.

---

## [Decision Letter · Decision Letter 1]

8 Jun 2025

Dear Dr. Guo,

Thank you for submitting your manuscript to PLOS ONE. After careful consideration, we feel that it has merit but does not fully meet PLOS ONE’s publication criteria as it currently stands. Therefore, we invite you to submit a revised version of the manuscript that addresses the points raised during the review process.

We look forward to receiving your revised manuscript.

Kind regards,

Mahdi Roozbeh

Academic Editor

PLOS ONE

Journal Requirements:

Reviewers' comments:

Reviewer's Responses to Questions

**Comments to the Author**

Reviewer #1: All comments have been addressed

Reviewer #2: (No Response)

2. Is the manuscript technically sound, and do the data support the conclusions?

Reviewer #1: Yes

Reviewer #2: Partly

3. Has the statistical analysis been performed appropriately and rigorously?

Reviewer #1: Yes

Reviewer #2: Yes

4. Have the authors made all data underlying the findings in their manuscript fully available?

Reviewer #1: Yes

Reviewer #2: Yes

5. Is the manuscript presented in an intelligible fashion and written in standard English?

Reviewer #1: Yes

Reviewer #2: Yes

Reviewer #1: All reviewer comments have been adequately addressed, and I am satisfied with the author's responses.

Reviewer #2: The authors have made efforts to address previous comments; however, I still recommend that they clarify the following key points to improve the clarity of their findings:

1. Please clarify whether p-values reported for enrichment analyses, correlation studies, and other statistical tests were adjusted for multiple comparisons (e.g., using FDR or Bonferroni correction).

2. While the construction of ceRNA networks for USP32 and ZNF655 is a potentially valuable addition, the analysis lacks prior introduction or justification for the relevance of miRNA/lncRNA in pulmonary hypertension. Please add a brief explanation and provide the methodological details for this analysis.

3. As noted previously, the discussion is overly broad and lacks a cohesive focus. It attempts to associate USP32 and ZNF655 with numerous biological pathways and mechanisms, many of which are tangential or speculative, and dilute the impact of their findings. Consider to streamline the discussion to emphasize a few well-supported, relevant pathways that directly connect their findings to pulmonary hypertension. This would be especially important given that there is no experimental validation of the involvement of USP32 or ZNF655 in pulmonary hypertension in this paper.

Other comments:

Figure 4G would be more informative if it contained the gene names instead of numbers.

Figures 6, 7a and 7c remain difficult to read. Please consider improving its resolution.

“Zinc lipoprotein (ZNF) family” appears to be a misstatement. Please revise for accuracy.

**Do you want your identity to be public for this peer review?** For information about this choice, including consent withdrawal, please see our Privacy Policy

Reviewer #1: No

Reviewer #2: No

---

## [Author Response · Author response to Decision Letter 2]

25 Jul 2025

Responses to Editor and Reviewer

Dear Academic Editor and Reviewer:

Hello

We are honored to receive your correspondence. We are grateful for the constructive comments and insightful recommendations provided on our manuscript, titled " Comprehensive analysis of disulfidptosis-related genes in pulmonary hypertension through machine learning and immune infiltration: spotlight on USP32 and ZNF655 as key regulators" We have meticulously revised the original manuscript and addressed each comment in detail. The following section will address each of these issues in turn. We are appreciative of the constructive feedback provided by the editors and reviewers, as it plays a pivotal role in enhancing the quality of our manuscripts. The following section presents a comprehensive, point-by-point response to all comments.

Academic Editor

We are very pleased to receive your positive feedback on our manuscript and your reminder for revisions. We have now completed the modifications to the manuscript as requested. In the revised version, we will submit it strictly according to your requirements. Thank you for your continued attention and help to our manuscript.

Reviewer #1

Thank you for your suggestions for improvements to our manuscript, and we are honored to have your suggestions along the way.

Reviewer #2

Comment 1:

Please clarify whether p-values reported for enrichment analyses, correlation studies, and other statistical tests were adjusted for multiple comparisons (e.g., using FDR or Bonferroni correction).

Response 1: It's great to receive your suggestions! Thank you for your attention to our manuscript. We clarify that the p-values reported in enrichment analyses, correlation studies, and other statistical tests are not corrected for multiple comparisons, so we did not specifically indicate this in the original text.

Comment 2:

While the construction of ceRNA networks for USP32 and ZNF655 is a potentially valuable addition, the analysis lacks prior introduction or justification for the relevance of miRNA/lncRNA in pulmonary hypertension. Please add a brief explanation and provide the methodological details for this analysis.

Response 2: It's great to receive your suggestions! Thank you for your attention to our manuscript. Based on your suggestion, we first added a description in the "Introduction" section, followed by a detailed methodological introduction in the "Methods and Results" section, which was also elaborated in the discussion.

Comment 3:

As noted previously, the discussion is overly broad and lacks a cohesive focus. It attempts to associate USP32 and ZNF655 with numerous biological pathways and mechanisms, many of which are tangential or speculative, and dilute the impact of their findings. Consider to streamline the discussion to emphasize a few well-supported, relevant pathways that directly connect their findings to pulmonary hypertension. This would be especially important given that there is no experimental validation of the involvement of USP32 or ZNF655 in pulmonary hypertension in this paper.

Response 3: It's great to receive your suggestions! Thank you for your attention to our manuscript. Based on your suggestions, we've overhauled and consolidated the original Tall & Thin discussion, highlighting a few of the issues you've highlighted.

Other comments

Comment 1:

Figure 4G would be more informative if it contained the gene names instead of numbers.

Response 1: It's great to receive your suggestions! Thank you for your attention to our manuscript. We have revised the content of Figure 4G.

Comment 2:

Figures 6, 7a and 7c remain difficult to read. Please consider improving its resolution.

Response2�It's great to receive your suggestions! Thank you for your attention to our manuscript. We've added clarity even further. To ensure clarity, we adjusted the clarity of these figures to 400dpi.

Comment 3:

“Zinc lipoprotein (ZNF) family” appears to be a misstatement. Please revise for accuracy.

Response3�It's great to receive your suggestions! Thank you for your attention to our manuscript. We're sorry for the inconvenience caused by our mistake, but our team took your suggestion very seriously, and we re-read the literature and found that the original text says "Zinc-finger proteins".The following is the relevant original image.

The following responses are intended to address the comments that have been submitted. We would like to express our gratitude once again for the time and effort that has been invested in the review of our work.

We extend our best wishes for the success of your future endeavors.

Sincerely,

Dingyu Guo

The Original Submitting Author

E-mail: 1743416277@qq.com

---

## [Decision Letter · Decision Letter 2]

7 Aug 2025

Comprehensive analysis of disulfidptosis-related genes in pulmonary hypertension through machine learning and immune infiltration: spotlight on USP32 and ZNF655 as key regulators

PONE-D-25-07068R2

Dear Dr. Dingyu Guo,

We’re pleased to inform you that your manuscript has been judged scientifically suitable for publication and will be formally accepted for publication once it meets all outstanding technical requirements.

Kind regards,

Mahdi Roozbeh

Academic Editor

PLOS ONE

**Comments to the Author**

Reviewer #2: All comments have been addressed

2. Is the manuscript technically sound, and do the data support the conclusions?

Reviewer #2: Yes

3. Has the statistical analysis been performed appropriately and rigorously?

Reviewer #2: Yes

4. Have the authors made all data underlying the findings in their manuscript fully available?

Reviewer #2: Yes

5. Is the manuscript presented in an intelligible fashion and written in standard English?

Reviewer #2: Yes

Reviewer #2: The authors' response addresses all previous concerns and suggestions - particularly, the discussion is now much more focused and objective.

**Do you want your identity to be public for this peer review?** For information about this choice, including consent withdrawal, please see our Privacy Policy

Reviewer #2: No

---

## [Editor Report · Acceptance letter]

PONE-D-25-07068R2

PLOS ONE

Dear Dr. Guo,

I'm pleased to inform you that your manuscript has been deemed suitable for publication in PLOS ONE. Congratulations! Your manuscript is now being handed over to our production team.

Kind regards,

on behalf of

Dr. Mahdi Roozbeh

Academic Editor

PLOS ONE